# “Transitivity”: A Code for Computing Kinetic and Related Parameters in Chemical Transformations and Transport Phenomena

**DOI:** 10.3390/molecules24193478

**Published:** 2019-09-25

**Authors:** Hugo G. Machado, Flávio O. Sanches-Neto, Nayara D. Coutinho, Kleber C. Mundim, Federico Palazzetti, Valter H. Carvalho-Silva

**Affiliations:** 1Grupo de Química Teórica e Estrutural de Anápolis, Centro de Pesquisa e Pós-Graduação. Universidade Estadual de Goiás, 75132-400 Anápolis, GO, Brazil; 2Instituto de Química, Universidade de Brasília, Caixa Postal 4478, 70904-970 Brasília, Brazil; kcmundim@unb.br; 3Dipartimento di Chimica, Biologia e Biotecnologie, Università di Perugia, 06123 Perugia, Italy

**Keywords:** *d*-TST, activation energy, Transitivity plot, solution kinetic

## Abstract

The Transitivity function, defined in terms of the reciprocal of the apparent activation energy, measures the propensity for a reaction to proceed and can provide a tool for implementing phenomenological kinetic models. Applications to systems which deviate from the Arrhenius law at low temperature encouraged the development of a user-friendly graphical interface for estimating the kinetic and thermodynamic parameters of physical and chemical processes. Here, we document the Transitivity code, written in Python, a free open-source code compatible with Windows, Linux and macOS platforms. Procedures are made available to evaluate the phenomenology of the temperature dependence of rate constants for processes from the Arrhenius and Transitivity plots. Reaction rate constants can be calculated by the traditional Transition-State Theory using a set of one-dimensional tunneling corrections (Bell (1935), Bell (1958), Skodje and Truhlar and, in particular, the deformed (d-TST) approach). To account for the solvent effect on reaction rate constant, implementation is given of the Kramers and of Collins–Kimball formulations. An input file generator is provided to run various molecular dynamics approaches in CPMD code. Examples are worked out and made available for testing. The novelty of this code is its general scope and particular exploit of d*-*formulations to cope with non-Arrhenius behavior at low temperatures, a topic which is the focus of recent intense investigations. We expect that this code serves as a quick and practical tool for data documentation from electronic structure calculations: It presents a very intuitive graphical interface which we believe to provide an excellent working tool for researchers and as courseware to teach statistical thermodynamics, thermochemistry, kinetics, and related areas.

## 1. Introduction

Recent applications of chemical kinetics to a variety of complex systems involves the accurate dealing of properties to be described by techniques, which treat a series of processes beyond elementary chemical quantum dynamics or even approximate classical and semiclassical approaches. We analyze in a companion paper [1] the state of our approaches to these problems from a general viewpoint. Here, we deal with explicit computational calculations that allow moving directly in a simple way to global applications. 

Recourse needs to be made at one stage or another to statistical treatments [1,2,3] among which there is modern insurgence with respect to more traditional ones—exemplary is, in particular, this topical collection essentially dedicated to “Fuzzy Logic” [4,5]. 

Information on the kinetic and related parameters in chemical transformations and transport phenomena and their role in complex mechanisms is needed: Particularly the temperature dependence of rate processes, k(T) and often in the low temperature range, where deviations from linearity of Arrhenius plots are revealed. The phenomenology of curvatures in Arrhenius plot span all of chemistry: From the long list that is continuously updated, we refer here to some selected cases, such as combustion chemistry [6], condensed-phase [7], atmospheric and astrochemical reactions [8,9], processes involved in the preservation and aging of food and drugs [10,11], as well as in basic geochemical [12] and biochemical environments [13,14]. The current status of the phenomenology is classified emphasizing case studies, specifically (i) super-Arrhenius kinetics, convex curvature in the Arrhenius plot, where transport phenomena brakes the processes as temperature decreases; (ii) sub-Arrhenius kinetics, concave curvature in the Arrhenius plot, where quantum mechanical tunneling propitiates low temperature reactivity; (iii) anti-Arrhenius kinetics, negative apparent activation energy, where processes are limited by stereodynamic requirements.

The curvature in the Arrhenius plot promotes a temperature dependence on the apparent activation energy $E_{a}$, the definition of which has been recommended by the International Union for Pure and Applied Chemistry (IUPAC) [13,15] as follows,
(1)Ea=−kBdlnk(T)d (1T)=−dlnk(β)d β.

In Equation (1), $k_{B}$ is the Boltzmann constant, T is the absolute temperature and $\beta =1/k_{B}T$. The so-called Tolman’s Theorem [16] proposed an interpretation for $E_{a}$ as the difference between the average over the energy of all reacting systems and energy of all (reactive and nonreactive) systems— propitiating a connection between canonical quantities with microcanonical features of the potential energy surface. Independently, the same spirit lead to transition-state formulations [15,16,17,18]. Recently, it was found expedient to introduce the reciprocal of the apparent activation energy in order to define the Transitivity function [2,3], Equation (2),
(2)$$\gamma \left(\beta \right)\equiv \frac{1}{E_{a}\left(\beta \right)}$$
to construct an appropriate scaling plane, γ(β) vs. β, where the regular curvatures on the Arrhenius plane are approximately linearized: The Transitivity plot. The function γ(β) can be interpreted as a measure of the propensity for the reaction to proceed and permit uniformly to account for experimental and theoretical rate processes, such as quantum tunneling, transport properties, and diffusion in the neighborhoods of phase transitions.

For cases where experimental and theoretical rate processes are difficult to interpret—by molecular complexity, extreme conditions or impossibility of exact solution to the Schrodinger’s equation—phenomenological and semiclassical theoretical approaches are of increasing utility. Over the years, the paramount useful phenomenological tool for studying the kinetics of physicochemical processes has been the Arrhenius law. It fails when the temperature range of the rate process becomes large: Empirical laws have been proposed, involving empirical parameters often lacking of physical interpretation, e.g., Kooij [19], power law [20], Vogel–Fulcher–Tamman (VFT) [21,22,23], Nakamura–Takayanagi–Sato (NTS) [24], deformed (d) Aquilanti–Mundim (AM) [25] and Aquilanti–Sanchez–Coutinho–Carvalho (ASCC) [26]. In Reference [27], it was presented how the transitivity concept can sort and interpret all these empirical laws, permitting a microscopic interpretation of the phenomenological parameters. 

For understanding and predicting a wide variety of kinetic processes with large molecular complexity and presenting deviation from Arrhenius law, Transition-State Theory (TST) remains an excellent formulation. The TST triggered the development of a variety of improved approaches: Important variants include the variational TST [28], the Marcus theory of electron transfer [29], and quantum [30] and path integral [31,32] versions. However, the variants of TST require additional information of the potential energy surface rendering formidable the task of estimating kinetic parameters as the size of molecular systems increases: Examples are the degradation kinetics of organic pollutants in aquatic, soil, and atmospheric environments [33,34,35,36,37,38]. 

The late twentieth century saw the advent, beyond the Transition State formulation, of molecular dynamics simulations, which is nowadays a powerful theoretical tool in understanding mechanisms of physical and chemical processes. Recently, there is a vast activity evaluating whether advances in molecular dynamics simulations can provide quantitatively rate constants [39,40,41]. However, the extraction of quantitative information on rate constants from molecular dynamics simulations is an important issue but a very difficult one to tackle [42,43,44].

Software tools to calculate and interpret rate constants are enormously useful in material, biochemical, and geochemical research to permit the exploitation of intense progress in computational hardware. Widely diverse codes have been developed to calculate rate constants in a gas-, liquid-, and solid-phase using Transition-State Theory and its variants: Polyrate [45], TheRate [46], MultiWell [47], TAMkin [48], Mesmer [49], RMG [50], APUAMA [51], KiSThelP [52], FRIGUS [53] and Eyringpy [54] codes are excellent options available to work out the kinetics of chemical reactions. Herein, we describe a new code with a user-friendly graphical interface able to perform various procedures for computing kinetic and related parameters in chemical transformations and transport phenomena: The code is referred to as “Transitivity” (see Figure 1), a concept related to the function defined in Equation (2); written in Python, it is a free open-source code compatible with Windows, Linux, and MacOS platforms. It offers the possibility of estimating phenomenological parameters from Arrhenius and Transitivity plots using a stochastic optimization method, Generalized Simulated Annealing (GSA) [55], with several options: Arrhenius, Aquilanti–Mundim (AM) [25], Vogel–Fulcher–Tammann (VFT) [22], Nakamura–Takayanagi–Sato (NTS) [24,56] and Aquilanti–Sanchez–Coutinho–Carvalho (ASCC) [26] formulas. This code also allows the estimation of unimolecular and bimolecular reaction rate constants with traditional TST using Bell35 [57], Bell58 [58], Skodje–Truhlar (ST) [59] and deformed (d-TST) corrections [60]. Solvent effects can be accounted for by the Collins–Kimball [61] and Kramers [62] models. Finally, input files for different first-principles molecular dynamics—Born–Oppenheimer Molecular Dynamics (BOMD), Car–Parrinello Molecular Dynamics (CPMD), Metadynamics, Path-Integral Molecular Dynamics (PIMD), and Trajectory Surface Hopping (TSH)—can be generated to run by the CPMD code [63]. 

We propose this code, as we expect it serves as a quick and practical tool for documentation data from electronic structure calculations. Additionally, it presents a very intuitive graphical interface which we believe to provide a useful working tool for the general public and researchers and also as courseware to teach statistical thermodynamics, thermochemistry, kinetics, and related areas. The body of this article is structured in two parts, one that deals with a brief theoretical and operational description of the code, and a second one that shows a variety of examples. A final section is devoted to additional and concluding remarks and an Appendix A collects used symbols and their meaning.

## 2. Theoretical Background

### 2.1. Phenomenology of Temperature Dependence of the Reaction Rate Constant

The theoretical apparatus to connect the Transitivity function, γ(T), and phenomenological reaction rate constant formulas and vice-versa is built and extensively discussed in Reference [27] and references therein. Classical and recent phenomenological reaction rate constant formulas to account for sub-, super-, and anti-Arrhenius behavior are applied in Section 4.1 by use of: (i) Deformed (d) Aquilanti–Mundim [25], (ii) VFT (Vogel–Fulcher–Tammann) [22], (iii) NTS (Nakamura–Takayanagi–Sato) [24,56], and (iv) ASCC (Aquilanti–Sanchez–Coutinho–Carvalho) [26]. More details of the formulas are presented in Table 1.

### 2.2. Calculation of Reaction Rate Constant

The Transition-State Theory, TST, is the most popular tool used to study the kinetics of chemical reactions with a well-defined activated complex—as customary, a double dagger (‡) denotes the properties pertaining to the transition-state complex. For a general bimolecular reaction, such as $\mathrm{Reactants}\;\to TS^{‡}\to \mathrm{Products},\;$it is necessary to compute the $Q_{1},\;\;\;Q_{2\;}$ and $Q^{‡}$ partition functions of reactants $R_{1}$ and $R_{2}$ and of the transition state, respectively. At absolute temperature T, the rate constant is given by: (3)$$k^{TST}\left(T\right)=\frac{k_{B}T}{h}\frac{Q^{‡}}{Q_{1}Q_{2\;}}\exp\left(-\frac{\varepsilon^{‡}}{k_{B}T}\right),$$
where h is Planck’s constant; and $\varepsilon^{‡}$ is the effective height of the energy barrier, eventually with the addition of the zero-point energy correction in Equation (3). The tunneling correction introduced in TST can be calculated by Skodje–Truhlar [59], Bell35 [57], and Bell58 [58] corrections and by the deformed (d) tunneling formulation presented in the next section [60].

#### 2.2.1. Deformed Transition-State Theory (d-TST)

The deformed Transition-State Theory (d-TST) [60] formulation:(4)$$k_{d}^{}\left(T\right)=\frac{k_{B}T}{h}\frac{Q^{‡}}{Q_{1}Q_{2}}{\left(1-d\frac{\varepsilon^{‡}}{k_{B}T}\right)}^{1/d},\;\;d=-\frac{1}{3}{\left(\frac{h\nu^{‡}}{2\varepsilon^{‡}}\right)}^{2},$$
where $\nu^{‡}$ is frequency for crossing the barrier, uniformly covers the range from classical to moderate tunneling regimes but needs an amendment for deep tunneling in exothermic reactions, a relatively rare case. The proposed variant of the transition-state theory was obtained from the transitivity concept and deformed (d) Aquilanti–Mundim law permit comparison with experiments and tests against alternative formulations. The nomenclature in Equation (4) is the same as used in Equation (3). A popular formulation in the literature is to use the Wigner tunneling correction: However, in our previous study [60,64], we have shown that d-TST is a more satisfactory approximation also in view of uniform behavior across the height of the energy barrier.

#### 2.2.2. Bell35 and Bell58

To cover the moderate-to-deep tunneling transition in exothermic reactions, we applied both Bell35 correction [57,65], Equation (5):(5)$$\kappa_{Bell35}=\;\frac{\left[\frac{1}{\hbar \nu^{‡}}-\;\frac{1}{k_{B}T}\exp\left(\frac{\varepsilon^{‡}}{k_{B}T}-\;\frac{\varepsilon^{‡}}{\hbar \nu^{‡}}\right)\right]}{\frac{1}{\hbar \nu^{‡}}-\;\frac{1}{k_{B}T}},$$
and Bell58 [58,65] correction truncated at the second term (*2T*)
(6)$$\kappa_{Bell58-2T}=\;\frac{\left(\frac{h\nu^{‡}}{2k_{B}T}\right)}{\sin\left(\frac{h\nu^{‡}}{2k_{B}T}\right)}-\;\frac{\exp\left(\frac{\varepsilon^{‡}}{k_{B}T}-\;\frac{\varepsilon^{‡}}{\hbar \nu^{‡}}\right)}{\left(\frac{k_{B}T}{\hbar \nu^{‡}}-\;1\;\right)},$$
that, although non-uniform across the transition between negligible and moderate tunneling regimes, were found to behave smoothly enough to adequately perform practically astride the whole range. 

Truncating Equation (6) to the first term, we recover the usual formula used in the literature to describe the tunneling under deep regime Equation (7):(7)$$\kappa_{Bell58}=\frac{\left(\frac{h\nu^{‡}}{2k_{B}T}\right)}{\sin\left(\frac{h\nu^{‡}}{2k_{B}T}\right)}$$

However, this formulation presents divergence at the crossover temperature $T_{c}=h\nu^{‡}/\pi k_{B}$. The tunneling regimes can be delimited within four temperatures ranges—negligible ($T>2T_{c}$), small ($2T_{c}>T>T_{c}$), moderate ($T_{c}>T>T_{c}/2$), and deep ($T>T_{c}/2$) [60,66,67].

#### 2.2.3. Skodje and Truhlar, ST

To avoid spurious divergence at $T_{c}$ in the Bell58 formulation, in 1981, Skodje and Truhlar [59] gave a generalization extending the parabolic barrier treatment. In their approximation for tunneling correction Equations (8a) and (8b)
(8a)$$\kappa_{ST}=\;\frac{\left(\frac{h\nu^{‡}}{2k_{B}T}\right)}{\sin\left(\frac{h\nu^{‡}}{2k_{B}T}\right)}-\;\frac{\exp\left[\left(\frac{1}{k_{B}T}-\;\frac{1}{\hbar \nu^{‡}}\right)\left(\varepsilon^{‡}-\Delta H\right)\right]}{\left(\frac{k_{B}T}{\hbar \nu^{‡}}-\;1\;\right)},\;\;\;\;\;\beta \leq 𝕔,$$
and
(8b)$$\kappa_{ST}=\;\frac{1}{\left(\frac{k_{B}T}{\hbar \nu^{‡}}-\;1\;\right)}\left\{\exp\left[\left(\frac{1}{k_{B}T}-\;\frac{1}{\hbar \nu^{‡}}\right)\left(\varepsilon^{‡}-\Delta H\right)\right]-1\right\},\;\;\;\;\;𝕔\leq \beta ,$$
where $𝕔=1/k_{B}T_{c}=\pi /h\nu^{‡}$ and ΔH is the enthalpy of reaction.

### 2.3. Solvent Effect on Reaction Rate Constant

#### 2.3.1. Collins–Kimball Formulation

Treatment of chemical reactions in liquid-phase requires accounting for the solvent effect, considering the ability of the reagents to diffuse and to lead them to effective reactive collisions. According to the Onsager solvent reaction field model [68,69,70], the solvent creates a solvation layer, as a cage around the molecular entities that participate in the reaction. The reactive process between molecular entities *A* and *B* is represented by the sequential equation,
(9)$$A_{solv}+\;B_{solv}\begin{array}{c} {\vec{k}}_{D}\\ ⇌\\ {\overset{\leftarrow }{k}}_{D} \end{array}{\left\{AB\right\}}_{solv}\overset{k}{\to }Product.$$

The “*solv”* symbol indicates that the molecular entity is surrounded by a roughly spherical solvation layer, the cage, having a specific radius. ${\vec{k}}_{D}$ is the diffusion kinetic rate constant which the reagent A travels in the solvent to find B; vice versa, ${\overset{\leftarrow }{k}}_{D}$ is the kinetic rate constant for reverse diffusion; k denotes the reaction rate constant due to effective reactive collisions: It can be estimated from Equations (3) or (4). Assuming the Steady-State Approximation [70] for Equation (9), the Smoluchowski expression for the diffusion kinetic rate constant [71], takes the form Equation (10):(10)$${\vec{k}}_{D}=4\pi r_{AB}D_{AB},$$
where $r_{AB}$ is a reaction radial distance and $D_{AB}$ is the sum of the diffusion constants for each reagent in the solvent. The generalization of Collins and Kimball [61,72] for the irreversible bimolecular diffusion-controlled reactions at infinite reaction rate, ${\vec{k}}_{D}≅{\overset{\leftarrow }{k}}_{D\;}$ [73], yields an overall reaction rate constant, $k_{Obs}$ Equation (11)
(11)$$\frac{1}{k_{Obs}}=\frac{1}{k_{TST}}+\frac{1}{{\vec{k}}_{D}}.$$

In the code, the diffusion is accounted for by the Stokes–Einstein formulation and the temperature dependence of viscosity η of the solvent is estimated through the Aquilanti–Mundim formula (see details in Section 3).

#### 2.3.2. Kramers’ Formulation

To account for dynamical effects of the solvent in a reactive process and to generalize to unimolecular and pseudo-unimolecular processes, Kramers’ model considers a stochastic motion of the system, where the solvent effect is added considering Brownian movements along the reaction path [70]. 

Assuming that the friction constant, μ (see below), is independent of time, the overall reaction rate constant $k_{Obs}$ can be calculated as Equation (12)
(12)$$k_{Obs}=\kappa_{Kr}k_{TST},$$
where $\kappa_{Kr}$ is the transmission factor obtained by Kramers [62] as:(13)$$\kappa_{Kr}=\frac{1}{\omega^{‡}}\left(\sqrt{\frac{\mu^{2}}{4}+\omega^{‡}{}^{2}}-\frac{\mu }{2}\right).$$

For the transition-state theory rate constant $k_{TST}$ and variants see Section 2.2. In Equation (13), $\omega^{‡}$ is the imaginary frequency of the transition state and the friction constant is given by $\mu =\left(6\pi r_{AB}/M\right)\eta$, where $r_{AB}$ and M are the radius of the cage and the molecular mass of the transition-state, respectively. Again, the viscosity is calculated in the code through the Aquilanti–Mundim formula.

## 3. Handling the Transitivity Code

Several aspects regarding nomenclature and the theoretical background used in this article are presented in already cited references and in Reference [27]. Furthermore, the code with the manual, examples (see next section), and installation video can be freely downloaded in the www.vhcsgroup.com/transitivity web page.

In the main window of the code, the user can choose between three options: (i) Fitting reaction rate constant data as a function of temperature; (ii) predicting the reaction rate constant in the gas- and liquid-phase; and (iii) creating the input for the calculation of first-principles molecular dynamics by the CPMD code. If the option chosen is “Kinetic and Related Parameters”, the program needs the electronic structure output files provided by the Gaussian program for the structures of the reagents, of the transition state and of the products. If the system under study requires the calculation of the electronic energy using a specifically high level method, the energy values must be included separately in the box. The code provides the reaction rate constants for both unimolecular and bimolecular reactions and the users should indicate if the molecular entity corresponds to an atom or whether the molecule is linear or not. In a new window, the user can choose between TST or d-TST [60] and Bell35 [57], Bell58 [58], Skodje–Truhlar (ST) [59] tunneling corrections. In addition, the program provides a visualization of the Arrhenius plot with the possibility of including any experimental and/or theoretical data available for validation. In the same window, reaction properties (such as internal energy, enthalpy, Gibbs-free energy, barrier height, d parameter, imaginary frequency ($\nu^{‡}$), crossover temperature ($T_{c}$), and the parameter of the Skodje–Truhlar model) are exhibited. 

If the VOLUME keyword is used in the calculation input file of G09, the option including solvent effects is available through the Kramers and Collins–Kimball formulations. It is necessary to make the choice of solvent parameters for the estimation of the viscosity or use the default (water). The parameters for the viscosity estimation (η/Poise) of a solvent other than water should be those fitted by the Aquilanti–Mundim formula [25] with $\eta_{o}$ in Poise and ε in J·mol^−1^ and inserted in the “Solvent Type” option. The temperature dependence of viscosity of water using the experimental data [74] is expressed as $\eta \left(T\right)=2.7024.10^{-4}Poise{\left(1-213.0543/T\right)}^{-2.75634}$. When the Kramers formulation is chosen, the friction coefficient of the solvent ($\mu /s^{-1})$, the Kramers transmission coefficient, and the overall reaction rate constant ($k_{Obs}/{\mathrm{cm}}^{3}\cdot {\mathrm{mol}}^{-1}\cdot s^{-1}\;$or $s^{-1}$) are provided. If the Collins–Kimball formulation is selected, the overall reaction rate constant ($k_{Obs}/{\mathrm{cm}}^{3}\cdot {\mathrm{mol}}^{-1}.s^{-1}$), separate diffusion coefficients (${\mathrm{cm}}^{2}\cdot s^{-1}$) for the reactants and the Smoluchowski diffusion rate constant (${\vec{k}}_{D}/{\mathrm{cm}}^{3}\cdot {\mathrm{mol}}^{-1}\cdot s^{-1})$ are provided. 

With the "GSA Fitting" option, the values of the rate constants and temperatures will be needed. In the software, the fitting in the Arrhenius plot is implemented for the rate constant data using the Arrhenius, Aquilanti–Mundim, VFT, NTS, and ASCC formulas by the stochastic optimization algorithm GSA. It is possible to insert the guess parameters d, $\varepsilon^{‡}$, $E_{a}$, $E_{0}$, $E_{\nu }$, *B* and $T_{0}\;$(see formulas in Table 1). Furthermore, information of the fit are available in “Fit [FormulaName].dat” output file. Internal parameters of GSA can be also controlled [55]. The fitting in the Transitivity plot is implemented of the reaction rate constant data using only Arrhenius, Aquilanti–Mundim, and VFT. The Transitivity plot is calculated using numerical differentiation with the option “Preview” and if necessary the smoothing of the data can be applied with the option “Apply SG” enabling the Savitzky–Golay filter [75]. In addition, the program provides a visualization of the Arrhenius and Transitivity plots with the fitting model chosen step-by-step.

The present code also offers the possibility of creating input to first-principles molecular dynamics simulation by the CPMD computational code. After selecting the input file of the system to be studied, which contains the molecular geometry of the system in a specific extension (*.xyz, *.gjf, *.out and *.log), the user should choose the molecular dynamics method. In addition, the user has the option to choose the Density Functional Theory (DFT) functional, pseudopotential, the temperature, the charge, the simulation time, and the integration time step. The size of the simulation box can be changed in the “Lattices” section. Additionally, the Transitivity code generates another output file with a *.gjf extension, where the user can check if the geometry of the system to be simulated is correct.

## 4. Examples

### 4.1. Fitting Mode—Arrhenius and Transitivity Plots

To illustrate the use of most of functionalities and to validate the accuracy of the running of the Transitivity code, a fitting of the reaction rate constants in the Arrhenius plot is performed as a function of temperature for four different systems in different regimes: sub-Arrhenius, corresponding to both deep and moderate tunneling, super-Arrhenius, and anti-Arrhenius behavior. 

The first example concerns the keto–enol tautomerization of 2-(2′-hydroxy-4′-methylphenyl) benzoxazole (MeBO), a well-known case of a deep-tunneling regime [7]: For this case we performed a fitting using Arrhenius, Aquilanti–Mundim, Nakamura–Takayanagi–Sato and ASCC formulas. In order to present the results for the sub-Arrhenius behavior under the moderate-tunneling regime, we evaluated the OH + H_2_
→ H + H_2_O reaction [76] using the Arrhenius and Aquilanti–Mundim formulas. The third example is centered on the investigations that have revealed the super-Arrhenius behavior for the rates of the processes promoted by enzymatic catalysis [77,78,79]. Here, we regard the reaction rate constant of the hydride transfer between the substrate and NAD^+^, catalyzed by F147L, which exhibits a strong convex curvature in the temperature range of 5 to 65 °C [78]: A fitting is performed with the Arrhenius, Aquilanti–Mundim and VTF formulas. The final case is aimed at showing a case of anti-Arrhenius behavior, which is characterized by the decrease of the reaction rate constant with the increase in the temperature. The OH + HBr→ Br + H_2_O reaction is prototypical in studies of an example, both from a theoretical and an experimental point of view: It exhibits negative temperature dependence of the reaction rate constant [80,81,82]. Fitting is performed with the Arrhenius and the Aquilanti–Mundim formulas.

The fitting parameters and statistical analysis of the quality of the fits for all the systems are given in Table 1. The statistical measure $\chi^{2}\;$is used in tests to compare the quality of the formula with the reference. It is noteworthy that in a fitting process, in order to obtain parameters with a physical meaning, it is necessary to avoid compensation effects [83] exploiting any prior knowledge of the system, since the formulas with more than two parameters can lead to multiple solutions.

Figure 2 shows the experimental data and fitted formulas for each system in the Arrhenius plot. Both NTS and ASCC formulas were satisfactory for fitting sub-Arrhenius behavior under deep-tunneling regime, while Aquilanti–Mundim is satisfactory just within a specific range of temperature. Under moderate-tunneling regime, the Aquilanti–Mundim formula was an excellent option to describe experimental data for OH + H_2_ reaction. The VTF and Aquilanti–Mundim formulas seem adequate to fit super-Arrhenius behavior. The Aquilanti–Mundim parameters that were obtained for the OH + HBr reaction indicate that they provide an excellent option for the anti-Arrhenius behavior. As expected, the Arrhenius formula is clearly inadequate to account for deviations at low temperature for all the reactions presented.

Curvature is easily identified through the Arrhenius plot in non-Arrhenius processes; however, the application of the phenomenological formulas to fit this behavior can lead to multiple solutions that make the physical interpretation of the obtained parameters difficult. In Reference [27], we show that curvature in the Arrhenius plot can be linearized using the transitivity plot. Analogous definition to transitivity function were proposed at 1980s to evaluate glass transition of supercooled materials [85,86,87]. However, the definitions failed to propose a connection with the Tolman theorem, Equation (1), being important only as scaling tools for relaxation processes. Figure 3 shows the Transitivity plot for the temperature dependence of relaxation time of the propylene carbonate [86], where a linear behavior is observed for a certain temperature range as expected by the deformed Aquilanti–Mundim law. However, at 198 K a transition to another linear regime is observed—a break not perceived in the Arrhenius plot. With the linearization of the data in the transitivity plot, the fitting process becomes much simpler: In the first high temperature range the Aquilanti–Mundim parameters are $\frac{1}{\varepsilon^{‡}}=1.71\;K^{-1}$ and d= 0.32; for the second lower temperature range, $\frac{1}{\varepsilon^{‡}}=0.53K^{-1}$ and d= 0.08. The interpretation of these results will be published in a future paper considering a large variety of examples.

### 4.2. Reaction Rate Constants’ Mode 

The estimation of the rate constants for the OH + HCl → H_2_O + Cl reaction in the gas-phase was performed to validate TST and tunneling corrections, implemented in the Transitivity code. Furthermore, using the suggestion of the Eyringpy code [54], the NH_3_ + OH → NH_2_ + H_2_O reaction was selected to demonstrate the accuracy of the Collins–Kimball and Kramers models to estimate reaction rate constants in an aqueous solution. 

#### 4.2.1. The OH + HCl → H_2_O + Cl Reaction

The reaction rate constants for the reaction between hydroxyl radical and hydrogen chloride are only slightly dependent on temperature in the range 138–300 K, although, as the range increases further, a temperature dependence is observed. This strong concave curvature detected in the Arrhenius plot (sub-Arrhenius behavior) is very convincing evidence of the role of deep quantum tunneling for this reaction [88]. We employed the d-TST, Bell35, Bell58, and ST tunneling corrections to calculate the rate constants for the reaction of OH + HCl over a wide range of temperatures (200–2000 K). The electronic structure properties of the reactants, of the products, and of the transition state were calculated employing the MP2/aug-cc-pVDZ calculation level using Gaussian 09 [89]. A complete study of this reaction using our methodology can be found in Reference [44].

Figure 4 obtained by the Transitivity code shows the comparison between the calculated reaction rate constant and the experimental data [90]. No major differences were found using either the Bel58-2T and ST tunneling corrections. Conversely, as expected, a divergence is observed of the Bell58 formula at $T_{c}$, and d-TST does not describe the range of experimental data at low temperature, where the deep-tunneling regime becomes dominant: This confirms that its validity is limited to weak tunneling. Traditional TST is presented for comparative purposes and visualizes the need for corrections.

#### 4.2.2. The NH_3_ + OH → NH_2_ + H_2_O Reaction

The NH_3_ + OH → NH_2_ + H_2_O reaction permits to illustrate the accuracy of the methodology in the liquid-phase. Energies, geometries, and frequencies of stationary points were extracted at the same level of calculation used in the Eyringpy code [54]. 

The upper panels in Figure 5 show the temperature dependence of the overall reaction rate constant, $k_{Obs}$, for NH_3_ + OH → NH_2_ + H_2_O reaction from 273 to 4000 K using Kramers and Collins–Kimball models. The Smoluchowski diffusion rate constant ${\vec{k}}_{D}$, which evaluates the diffusion limit for a bimolecular reactive process including the solvent effect, is shown in the lower right panel. The Kramers transmission correction, which evaluates the interference of the friction effect of the solvent in the reactive process as a function of temperature, is shown in the lower left panel. At 298.15 K, Kramers’ formulation gives for the reaction rate constant the value 6.73 × 10^11^ cm^3^ mol^−1^ s^−1^, while the Collins-Kimball formulation yields 6.77 × 10^11^ cm^3^ mol^−1^ s^−1^, (the experimental value indicates ~10^11^ cm^3^ mol^−1^ s^−1^) [91,92,93]. The value of $\;{\vec{k}}_{D}$ from Smoluchowski (Collins–Kimball) is 3.73 × 10^12^ cm^3^ mol^−1^ s^−1^, in accordance with that calculated in Reference [54], ${\vec{k}}_{D}$ = 3.60 × 10^12^ cm^3^ mol^−1^ s^−1^.

### 4.3. CPMD Input Files Generator

Coordinates of specific molecules can be selected to test the function of creating a first-principles molecular dynamics input to run by the CPMD code (Figure 6). When selecting this option, the user must choose between the CPMD, PIMD, TSH, MTD and BOMD approaches and must provide the molecular geometry of the system in *.xyz format. Additional information (DFT functional, simulation temperature, system charge, the maximum number of steps, integration time step, and the size of the box) can be inserted through the indicated boxes. Furthermore, it is possible to generate several aleatory configurations by selecting specific initial conditions for the geometric parameters (For more information, see the input files in the Example directory on the www.vhcsgroup.com/transitivity web page).

## 5. Final Remarks

The Transitivity code presented in this article tends to systematize the tools developed over the last ten years [25] to handle the kinetics of processes beyond Arrhenius, providing options for: (i) The phenomenological fitting and estimation of reaction rate constants in gas- and liquid-phases and (ii) the preparation of first-principles molecular dynamics in order to evaluate the parameters relative to a variety of reactive processes. In addition, the code provides an easy user-friendly interface and may be relevant for didactic purposes.

A characterizing feature of the code is the consistent use of the d-formulation, which recently culminated in a series of successful applications from phenomenological to first-principles descriptions of pure and applied chemical kinetics and material science. Examples are available:From the phenomenology of elementary processes (such as the H_2_ + F [94], OH + HBr [80], F + HD [65] and C + CH^+^ [95] reactions) to complex processes (such as food systems [96], plant respiration [25], plasma chemistry [97], and solid-state diffusive reaction [98]);Calculation of the kinetic rate constants for chemical reactions from the potential energy surface features profile, such as the CH_4_ + OH [60], CH_3_OH + H [99], OH + HCl [44], OH + HI [43], to proton rearrangement of enol forms of curcumin [100], OH + H_2_ [101], and chiral nucleophilic substitution reaction [102].

Regarding the evolution and consistency of the code, efforts are being made to introduce d predictive formulations to transport properties, as well as post-processing of trajectories obtained by first-principles molecular dynamics simulations. Furthermore, the Eckart tunneling correction and the variational Transition-State Theory represent focuses for the future implementation of the Transitivity code.

## Figures and Tables

**Figure 1 molecules-24-03478-f001:**
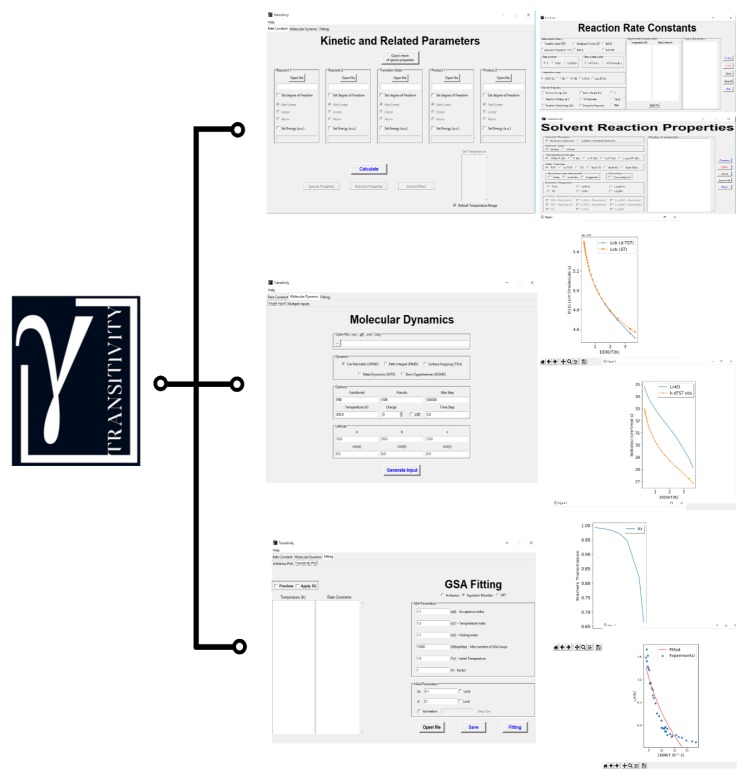
Logo and main windows of the Transitivity code.

**Figure 2 molecules-24-03478-f002:**
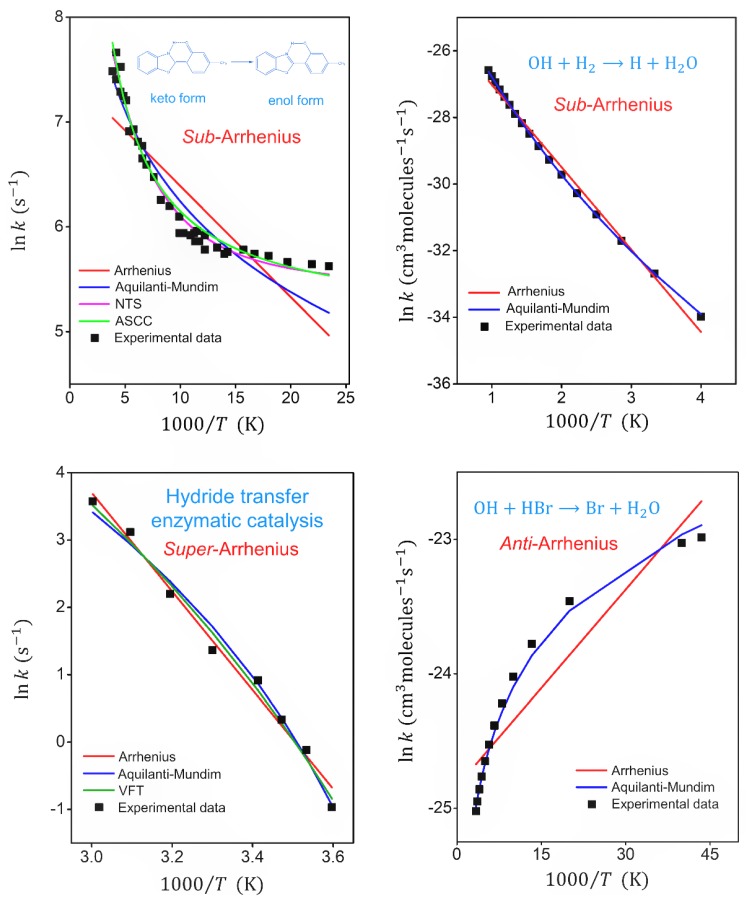
Arrhenius plots comparing the experimental reaction rate constant and fitted formulas for keto–enol tautomerization reaction (sub-Arrhenius behavior under deep tunneling), OH + H_2_ ⟶ H_2_O + H reaction (sub-Arrhenius behavior under moderate tunneling), hydride transfer with enzymatic catalysis (super-Arrhenius behavior) and OH + HBr ⟶ H_2_O + Br reaction (anti-Arrhenius behavior). NTS and ASCC formulas were of use for sub-Arrhenius behavior under deep-tunneling regime. The Aquilanti–Mundim formula was of use for sub-Arrhenius cases under moderate-tunneling regime, for super-Arrhenius and for anti-Arrhenius behaviors. VFT also was of use for super-Arrhenius situations. The references of experimental data can be found in Table 1.

**Figure 3 molecules-24-03478-f003:**
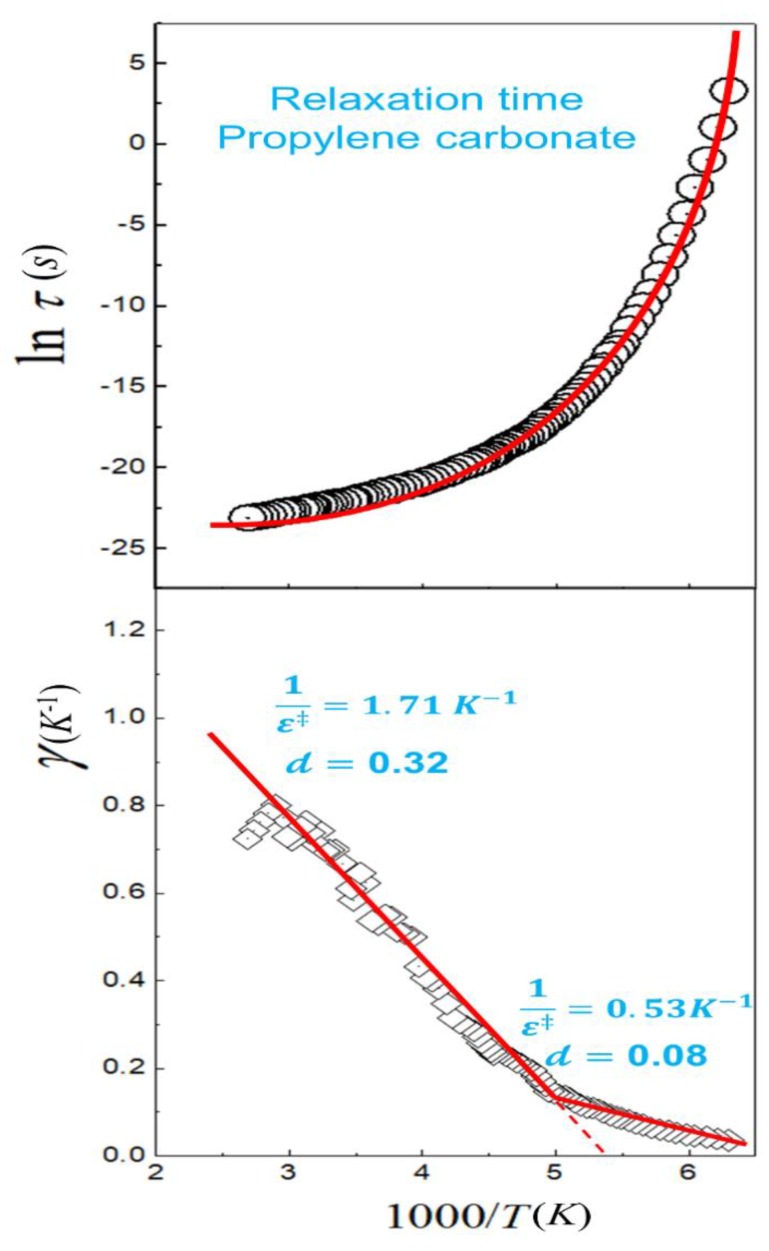
The Arrhenius (upper panel) and Transitivity (lower panel) planes of the temperature dependence of relaxation time of the propylene carbonate. The diamond symbols represent the transitivity values obtained numerically and smoothing with the Savitzky–Golay filter. Red lines emphasize two regions where the temperature dependence of the transitivity is linearized, as expected by the Aquilanti–Mundim law.

**Figure 4 molecules-24-03478-f004:**
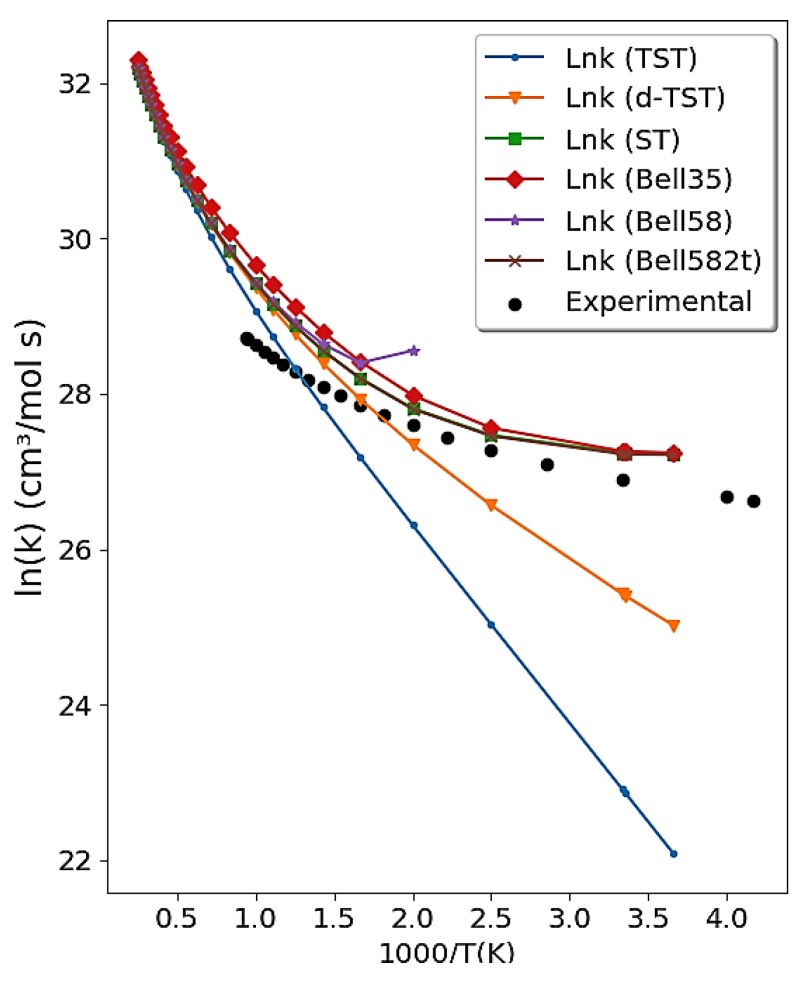
Arrhenius plot obtained from the Transitivity code for the OH + HCl → Cl + H_2_O reaction using TST with Bell35, Bell58, ST tunneling correction, and d-TST. Experimental data in the literature [90] are available for comparison and shown as full dots.

**Figure 5 molecules-24-03478-f005:**
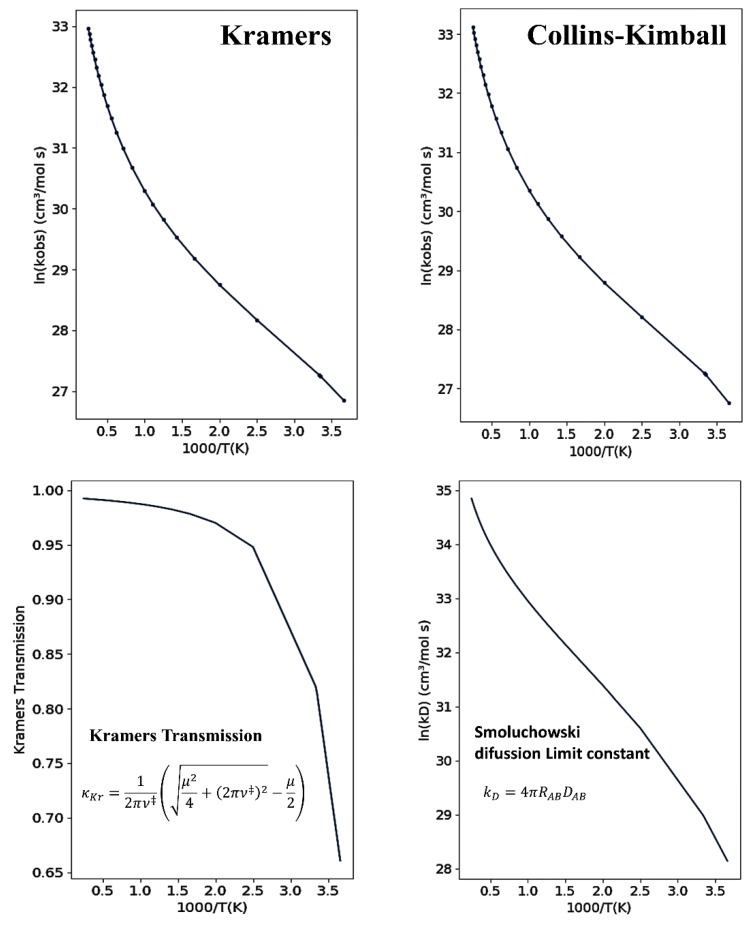
Upper panels present the Arrhenius plots as given by the program for the NH_3_ + OH → NH_2_ + H_2_O reaction using Kramers’ and Collins–Kimball formulations. The lower panels show the Kramers transmission and Smoluchowski diffusion limit constant as a function of inverse temperature.

**Figure 6 molecules-24-03478-f006:**
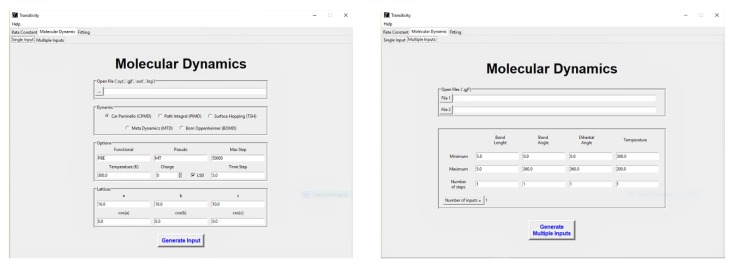
An exemplary view of the input generation function of the transitivity program. Details of input files can be found in the www.vhcsgroup.com/transitivity web page.

**Table 1 molecules-24-03478-t001:** Fitted parameters for the Arrhenius, AM, ASCC, NTS and VFT formulas, using the Transitivity code for keto–enol tautomerization [7], OH + H_2_ [76], enzymatic catalysis [78] and OH + Br [84] reactions. Energy ($E_{a}$, $\varepsilon^{‡}$, $E_{\upsilon }$ and $E_{0}$) is in cal/mol and temperature ($T_{0}\;$and B) in K. Pre-factor units can be identified in the references.

Formula		Chemical Processes			
	Fitted Parameters	Keto-enol Tautomerization [7] *Sub* -Arrhenius (Deep-Tunneling)	OH + H_2_ → H + H_2_ [76] *Sub*-Arrhenius (Moderate Tunneling)	Enzymatic Catalysis [78] *Super*-Arrhenius	OH + HBr → Br + H_2_O [84] *Anti*-Arrhenius
Arrhenius $k\left(T\right)=A\exp\left(-\frac{E_{a}}{k_{B}T}\right)$	A	1.74 × 10^3^	2.16·10^-11^	1.52 × 10^11^	1.66 × 10^-11^
	$E_{a}$	214	4891	14600	−94.6
	χ2	1.10 × 10^-2^	4.20·10^-3^	2.60 × 10^-2^	6.69 × 10^-2^
Aquilanti–Mundim (AM) k(T)=A(1−dε‡kBT)1d	A	3.32 × 10^6^	1.11·10^-10^	1.91 × 10^4^	7.43 × 10^-14^
	$\varepsilon^{‡}$	318.06	9170	2391	−324.61
	d	−0.81	−0.086	0.207	1.24
	χ2	3.68 × 10^-2^	6.80·10^-4^	2.91 × 10^-2^	2.78 × 10^-3^
Aquilanti–Sanchez–Coutinho–Carvalho (ASCC) $k\left(T\right)=A{\left(1-d\frac{\varepsilon^{‡}}{k_{B}T+E_{\nu }}\right)}^{\frac{1}{d}}$, $\;d=-\frac{1}{3}{\left(\frac{E_{\upsilon }}{2\varepsilon^{‡}}\right)}^{2}$	A	2.33 × 10^4^	-	-	-
	$\varepsilon^{‡}$	2441	-	-	-
	$E_{\upsilon }$	429	-	-	-
	χ2	2.18 × 10^-2^	-	-	-
Sato–Nakamura–Takayanagi (NTS) k(T)=Aexp[−E0kB(T2+T02)12]	A	3.12 × 10^4^	-	-	-
	$E_{0}$	1655	-	-	-
	$T_{0}$	168	-	-	-
	χ2	7.38 × 10^-3^	-	-	-
Vogel–Fulcher–Tammann (VFT) $k\left(T\right)=A\exp\left(\frac{B}{T-T_{0}}\right)$	A	-	-	1.25 × 10^5^	-
	B	-	-	−1298	-
	$T_{0}$	-	-	175	-
	χ2	-	-	2.16 × 10^-2^	-

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
