# Peer review of "“Transitivity”: A Code for Computing Kinetic and Related Parameters in Chemical Transformations and Transport Phenomena"

_molecules, 2019, doi:10.3390/molecules24193478_

Round 1
Reviewer 1 Report
In this article, the software “Transitivity” for computing thermodynamic, kinetic and related parameters in chemical transformations and transport phenomena is presented. The code has been written in PYTHON with a suitable user graphical interface. The theoretical background and the implemented equations are briefly described. However, the authors give an extended literature of their own published work as well as of the other pioneers. The present software adds new advances in fitting Arrhenius plots (sub-, super-, anti-) compared to previous software offered to the researchers in the field of chemical kinetics. Also, “Transitivity” provides tools (input files) for interfacing with first-principle molecular dynamics codes like CPMD and reading data from the GAUSSIAN package of codes.
This article is not a research paper and I do not know if MOLECULES is the appropriate journal for publishing this type of work. It is well known that journals for publishing scientific software are available. Nevertheless, I have the following remarks on the manuscript.
Because of the plethora of terms and symbols used, I wonder if a table, which provides a glossary of terms and symbols would be useful. The authors may consider this. The Appendix provides a brief description of well-known molecular dynamics methods and definitely is not suitable as it is for teaching the methods. I do not think that this appendix is necessary and simply giving the definitions of the methods with the relevant references will be adequate. Figure 4 is not very useful as is shown, since one cannot read the details of the input windows. It would be better if the graphical user interface windows are shown in more than one figures. English need care. I have written down a few grammatical corrections but I am sure more exist.The abstract should not contain abbreviations, which are introduced and explained later on, such as Bell35, Bell58, Skodje-Truhlar … (?) approach. In Introduction Line 87 … It fails … Line 94 … of kinetic processes … Line 103 … which is … Line 109 … exploitation … Line 131 Additionally … Line 198 … to approach and find ? … Line 232 ... high level method ... Lines 363 and 364, reverse … left(right) panel … Reference 62 … Physica
Author Response
In this article, the software “Transitivity” for computing thermodynamic, kinetic and related parameters in chemical transformations and transport phenomena is presented. The code has been written in PYTHON with a suitable user graphical interface. The theoretical background and the implemented equations are briefly described. However, the authors give an extended literature of their own published work as well as of the other pioneers. The present software adds new advances in fitting Arrhenius plots (sub-, super-, anti-) compared to previous software offered to the researchers in the field of chemical kinetics. Also, “Transitivity” provides tools (input files) for interfacing with first-principle molecular dynamics codes like CPMD and reading data from the GAUSSIAN package of codes.
This article is not a research paper and I do not know if MOLECULES is the appropriate journal for publishing this type of work.
ANSWER: We hope that editors agree and believe so especially in connection with the twin paper in Ref. [3] and the spirit of the special issue
It is well known that journals for publishing scientific software are available. Nevertheless, I have the following remarks on the manuscript.
Because of the plethora of terms and symbols used, I wonder if a table, which provides a glossary of terms and symbols would be useful. The authors may consider this.
ANSWER: Done.
The Appendix provides a brief description of well-known molecular dynamics methods and definitely is not suitable as it is for teaching the methods. I do not think that this appendix is necessary and simply giving the definitions of the methods with the relevant references will be adequate.
ANSWER: Done
Figure 4 is not very useful as is shown, since one cannot read the details of the input windows. It would be better if the graphical user interface windows are shown in more than one figures.
ANSWER: Done
English need care. I have written down a few grammatical corrections but I am sure more exist.
The abstract should not contain abbreviations, which are introduced and explained later on, such as Bell35, Bell58, Skodje-Truhlar … (?) approach. In Introduction Line 87 … It fails … Line 94 … of kinetic processes … Line 103 … which is … Line 109 … exploitation … Line 131 Additionally … Line 198 … to approach and find ? … Line 232 ... high level method ... Lines 363 and 364, reverse … left(right) panel … Reference 62 … Physica
ANSWER: Done
Reviewer 2 Report
This manuscript present software to estimate kinetic and thermodynamic parameters using a user-friendly user interface. It is interesting the fact that is full compatible with different OS and that is a open source code. The authors use different examples to text the software efficiency and they demonstrate that it would be a valuable tool both in research and in teaching. I can recommend its publication
Author Response
Thanks.

Round 2
Reviewer 1 Report
The new version has considerably been improved. Provided the journal accepts this technical type of paper the article can be published.